# Optimization of the Flavonoid Extraction Process from the Stem and Leaves of Epimedium Brevicornum and Its Effects on Cyclophosphamide-Induced Renal Injury

**DOI:** 10.3390/molecules29010207

**Published:** 2023-12-29

**Authors:** Meiling Shi, Hongyan Pei, Li Sun, Weijia Chen, Ying Zong, Yan Zhao, Rui Du, Zhongmei He

**Affiliations:** 1College of Traditional Chinese Medicine, Jilin Agricultural University, Changchun 130118, China; anxia143341@163.com (M.S.); phy19990505@163.com (H.P.); 20221595@mails.jlau.edu.cn (L.S.); chenweijia_jlau@163.com (W.C.); zongying@jlau.edu.cn (Y.Z.); zhaoyan@jlau.edu.cn (Y.Z.); durui@jlau.edu.cn (R.D.); 2Engineering Research Center for Efficient Breeding and Product Development of Sika Deer, Changchun 130118, China

**Keywords:** response surface optimization extraction, Epimedium flavonoids, cyclophosphamide, antioxidant and anti-inflammatory, kidney injury, molecular docking, Icariin

## Abstract

Cyclophosphamide (CTX) is a broad-spectrum alkylated antitumor drug. It is clinically used in the treatment of a variety of cancers, and renal toxicity is one of the adverse reactions after long-term or repeated use, which not only limits the therapeutic effect of CTX, but also increases the probability of kidney lesions. The total flavonoids of Epimedium stem and leaf (EBF) and Icariin (ICA) are the main medicinal components of Epimedium, and ICA is one of the main active substances in EBF. Modern pharmacological studies have shown that EBF has a variety of biological activities such as improving osteoporosis, promoting cell proliferation, antioxidant and anti-inflammatory properties, etc. However, few studies have been conducted on the nephrotoxicity caused by optimized CTX extraction, and protein-ligand binding has not been involved. This research, through the response surface optimization extraction of EBF, obtained the best extraction conditions: ethanol concentration was 60%, solid-liquid ratio of 25:1, ultrasonic time was about 25 min. Combined with mass spectrometry (MS) analysis, EBF contained ICA, ichopidin A, ichopidin B, ichopidin C, and other components. In this study, we adopted a computational chemistry method called molecular docking, and the results show that Icariin was well bound to the antioxidant target proteins KEAP1 and NRF2, and the anti-inflammatory target proteins COX-2 and NF-κB, with free binding energies of −9.8 kcal/mol, −11.0 kcal/mol, −10.0 kcal/mol, and −8.1 kcal/mol, respectively. To study the protective effect of EBF on the nephrotoxicity of CTX, 40 male Kunming mice (weight 18 ± 22) were injected with CTX (80 mg/kg) for 7 days to establish the nephrotoxicity model and were treated with EBF (50 mg/kg, 100 mg/kg) for 8 days by gavage. After CTX administration, MDA, BUN, Cre, and IL-6 levels in serum increased, MDA increased in kidney, GPT/ALT and IL-6 increased in liver, and IL-6 increased in spleen and was significant ((*p* < 0.05 or (*p* < 0.01)). Histopathological observation showed that renal cortex glomerular atrophy necrosis, medullary inflammatory cell infiltration, and other lesions. After administration of EBF, CTX-induced increase in serum level of related indexes was reduced, and MDA in kidney, GPT/ALT and IL-6 in liver, and IL-6 in spleen were increased. At the same time, histopathological findings showed that the necrosis of medullary and corticorenal tubular epithelium was relieved at EBF (50 mg/kg) dose compared with the CTX group, and the glomerular tubular necrosis gradually became normal at EBF (100 mg/kg) dose. Western blot analysis of Keap1 and Nrf2 protein expression in kidney tissue showed that compared with model CTX group, the drug administration group could alleviate the high expression of Keap1 protein and low expression of Nrf2 protein in kidney tissue. Conclusion: After the optimal extraction of total flavonoids from the stems and leaves of Epimedium, the molecular docking technique combined with animal experiments suggested that the effective component of the total flavonoids of Epimedium might activate the Keap1-Nrf2 signaling pathway after treatment to reduce the inflammation and oxidative stress of kidney tissue, so as to reduce kidney damage and improve kidney function. Therefore, EBF may become a new natural protective agent for CTX chemotherapy in the future.

## 1. Introduction

One of the key target organs for researching drug toxicity is the kidney. Due to the activation of polymorphic metabolic enzymes, CTX is a chemotherapy medication routinely used in clinics with significant adverse effects [1,2]. Approximately 30% of this substance is excreted in the urine in an activated state, causing severe nephrotoxicity and urinary system side effects in cystic fibrosis, as well as producing related inflammatory markers [3].

It poses a significant challenge to the daily life of human beings. Clinical manifestations of myelosuppression, toxic cystitis [4], nephrotoxic injury, and other diseases have a high mortality rate [5]. Nevertheless, the available natural medicines are few, costly, and have more substantial side effects, so they need to catch up with screening for natural drugs that have protective effects against CTX-induced kidney injury. Some natural products have been shown to alleviate CTX-induced nephrotoxicity during chemotherapy [6], but there is limited research on EBF.

Existing studies have shown that CTX toxicity leads to elevated alanine aminotransferase (ALT) and aspartate aminotransferase (AST) in serum and liver, and elevated levels of cytokines such as interleukin (IL)-1β, IL-6, and tumor elevation of necrosis factor-alpha in serum in mice [7]. Although the concept of nephrotoxicity has been recognized for more than 80 years, most studies are focused on pharmacological and medically induced injuries. In contrast, currently, there is a greater focus on raising awareness, avoiding nephrotoxins [8], mitigating early injury, and reducing disease severity. CTX is also frequently used in clinics as an alkylating agent to treat vasculitis [9]. This autoimmune disease leads to infections and malignant tumors damaging the lungs, causing kidney damage and multiple neuroinflammation [10], which can be associated with significant respiratory distress and segmental necrotizing glomerulonephritis with concomitant renal tubular atrophy and interstitial fibrosis [11]. Early toxicities due to CTX can be managed by stopping CTX immediately, while late toxicities are not fully recovered with treatment. As a broad-spectrum antineoplastic drug, its use is escalating as the global prevalence of cancer is increasing dramatically [12], according to IMS Health data. CTX is activating in the liver [13], and its metabolites have shown to have mutagenic-teratogenic solid effects [14], especially in organs and bones, and especially in pregnant women, where the toxicity of CTX can pass directly through the placenta barrier or indirectly penetrate the intrauterine and extrauterine environments [13], triggering embryonic malformations, stillbirths, and growth retardation. Therefore, it is imperative to investigate natural drug candidates with unique pharmacological effects and abundant natural active ingredients to alleviate the nephrotoxicity of CTX.

Epimedium (Epimedium brevicornu Maxim.) is a plant in the Berberidaceae family, and its main active ingredients are flavonoids, polysaccharides, and phenols. Recent studies have shown that Epimedium can be used as a natural drug candidate for improved treatment and prevention of diabetes mellitus [14]. The chronic hyperglycemia of diabetes leads to renal organ damage, renal dysfunction, and failure [15], especially of the nerves, kidneys, heart, and blood vessels, and pharmacological interventions aiming at Epimedium’s potential for preventive and therapeutic inhibition of α-glucosidase is a potential strategy for the treatment of diabetes-mellitus-induced renal damage and neurological damage [14]. Icariin is the main active ingredient in EBF. In the study of the effect of Icariin on cardiac function and mitochondrial oxidative stress in streptozotocin (STZ)-induced diabetic rats, it was found that its high-dose group significantly reduced the levels of cardiac mitochondrial MDA in the heart of diabetic-induced renal injury and nerve-injury-induced rats, as well as increased the activity of SOD [16]. In exploring the protective effect of Betulinic acid on cyclophosphamide-induced renal injury in mice, it was found that it could be involved in the protective effect by inhibiting the NF-κ B pathway and ERK-mediated mitochondrial apoptosis pathway [17]. Renal injury is the result of the progression of chronic renal failure [18], and EBF can exert anti-renal failure effects through activation of adenosine-monophosphate-activated protein kinase (AMPK) [19], redox regulation, cell growth, and proliferation, autophagy, and inflammation [19], among other functions [20]. EBF has been reported to have various pharmacological effects, such as anti-inflammatory [21], antioxidant [22], antibacterial [23], and antiviral [24,25]. Natural medicines are multi-targeted and can simultaneously modulate multiple biological pathways to achieve disease treatment and prevention [26,27]. Therefore, the present study is designed as a single-factor extraction, and the response-surface-optimized EBF extraction is already available for subsequent studies. Free radical scavenging is an integral part of normal metabolism in organisms, and a complex system of endogenous and exogenous sources of antioxidants in the body is used to mitigate the harmful effects of free radicals, an excess of which will lead to the development of diseases [28]. Tapping drugs that can effectively improve antioxidants is the first task in the fight against diseases [29].

Molecular docking is a new method based on computerized structures to predict ligand-target interactions at the molecular level [30]. The possibilities offered by molecular docking in drug discovery are well recognized [31,32], and ligand-based methods have been used to select suitable protein conformations for docking screening [33]. After the quantitative and qualitative analysis of the MS results, our laboratory used Icariin, an Icariin analog, for subsequent target docking.

There have been early studies on the protective effect of total flavones of Epimedium on the reproductive oxidative damage induced by CTX in male mice [34]. However, to our knowledge, there are few studies on the effect of total flavonoids of Epimedium stem and leaf on kidney injury caused by broad-spectrum anti-tumor drug CTX, and ligand-receptor binding was not included. Therefore, the purpose of this study was to optimize the extraction of total flavonoids from the stems and leaves of Epimedium by surface response, simulating the interaction between small ligand molecules and large receptor molecules by molecular docking technology, It was found that Icariin, an effective active component of EBF, was well docked with KEAP1, NRF2, COX-2, and NF-κB. Animal experiments showed that EBF could alleviate the increase in MDA level, the overexpression of Keap1 protein, and the low expression of NRF2 protein in kidney induced by CTX. The levels of BUN, Cre, IL-6, and MDA in serum with elevated CTX were decreased; also decreased were GPT/ALT and IL-6 levels in liver and IL-6 levels in spleen. HE and PAS sections also showed that EBF administration could improve glomerular atrophy, cell necrosis, and inflammatory infiltration in renal tissue caused by CTX. Among them, the high dose of 100 mg/kg EBF was more significant than the low dose of 50 mg/kg EBF, and the renal cortex and medullary tissue tended to be normal. It was speculated that Icariin, a natural product of the total flavonoids of Epimedium stems and leaves, may play a protective role in CTX-induced kidney injury through the Keap1-Nrf2 pathway.

## 2. Results and Discussion

### 2.1. Compound Preparation Results for Epimedium Stem and Leaf Flavonoids

#### 2.1.1. Standard Curves

Draw the standard curve according to the sample determination of Section 3.2.2 methods. The standard curve was prepared according to the standard curve preparation method, as shown in Figure 1A, and the absorbance was measured by the enzyme marker with a linear regression equation of y = 0.065x − 0.0313, R² = 0.9956.

#### 2.1.2. Results of the One-Way Experiment

In previous studies, the yields of ethanol extracts and n-butanol fractions were low [35]; there were few studies on class extraction through response surface optimization. The extraction of EBF under the best conditions is determined by increasing the original extraction rate. The effect of each extraction factor on the total flavonoid content under the condition of other factors remaining unchanged is shown in Figure 1. Figure 1B shows that as the volume fraction of ethanol changes, the yield of total flavonoids of Epimedium also changes, and it reaches the peak value at 60%. The material-liquid ratio was optimal at 25:1 (Figure 1C), and the peak was best reached by ultrasonication for 25 min (Figure 1D).

#### 2.1.3. Response Surface Experiment Results

The results of the response surface experiments are shown in the following table and Figure 2.

ANOVA and its test of significance was performed, with the yield of total flavonoids of Cynodon grandifloras as the response value. A quadratic polynomial stepwise regression was fitted to the experimental data of Table 1 using the Design Expert software 8.0.6, and the mathematical model obtained was Y = 44.08 + 0.55X1 + 1.28X2 + 0.25X3 − 0.098X1X2 − 0.155X1X3 − 0.1X2X3 − 1.04x12 − 1.49x22 − 0.59x32. The results of the significance test and ANOVA of regression equations are shown in Table 2.

The response surface method was used to optimize the process conditions of ultrasonic extraction of total flavones from chicken schisandra chinensis, and the anti-inflammatory effects of total flavones from chicken schisandra chinensis in vitro and in vivo were studied [36]. The response surface and contour map of Table 2 and Figure 2 were analyzed, and it was found that ethanol concentration, solid-liquid ratio, and ultrasonic time had significant effects on flavonoid content (*p* < 0.05).

According to the response surface design in the experimental method. In this experiment, ethanol concentration, solid-liquid ratio, and ultrasonic treatment time were selected as three important influencing factors, other influencing factors were fixed, and experimental design was carried out. 

#### 2.1.4. Validation Results of Optimal Extraction Conditions

According to the regression equation model, the optimal conditions obtained are ethanol concentration of 60%, material-liquid ratio of 25:1, and ultrasonic time of 25 min.

### 2.2. Mass Spectrometry Analysis of the Flavonoid Composition and Content of Epimedium Stems and Leaves

The flavonoids in Epimedium are the main components of Epimedium, including Epimedoside, Epimedosin, and Chaohuodin A–C. The total flavonoids in Epimedium were extracted and purified using the response surface method. In our laboratory, response surface methodology was implemented to optimize the total flavonoids extraction conditions of Epimedium extracted and identified after purification of the mass spectrometry test results. Table 3 shows the detection and analysis of a variety of compounds, including Epimedium glycoside, Asiaticoside A, Asiaticoside B, Asiaticoside C, Bao Huo Glycoside, Asiaticoside HypoglycosideII, Dehydrated Epimedicin, Anhydrocicaritin, and other chemical constituents. Studies have shown that 379 compounds were detected in the total flavonoids of Epimedium, including flavonoids, lignans, organic acids, terpenoids, dihydrophenanthrene derivatives, alkaloids, etc. [37]. Icariin, icaridine A, icaridine B, and icaridine C have been recognized as the main phytochemical and pharmacological active components of flavonoids.

The total flavonoids of Epimedium stem and leaves were extracted after one-way analysis of response surface optimized extraction conditions, identified and purified, and then analyzed by MS to obtain peaks in the positive ion mode (Figure 3A,B) as well as the peaks in the negative ion mode (Figure 3C,D). The ability of EBF to scavenge DPPH free radicals is shown in Figure 3E, and the ability of EBF to scavenge ABTS free radicals is shown in Figure 3F.

### 2.3. In Vitro Antioxidant Activity Assay Results

DPPH free radical scavenging activity is widely used to evaluate the antioxidant capacity of biological samples. It is a stable free radical that exhibits the strongest absorption at 517 nm. In this method, antioxidant substances can reduce DPPH free radicals to yellow diphenylpicrohydrazine [38]. The antioxidant capacity of the sample was linear and dose-dependent with the content of yellow compound in the reaction system. Therefore, the antioxidant activity of the sample can be expressed as its ability to clear DPPH free radicals. The ability of EBF to scavenge DPPH free radicals is shown in Figure 3C. The results showed that the total flavonoids and vitamin C of Epimedium enhanced DPPH free radical scavenging activity in a concentration-dependent manner. As the EBF concentration increases from 0.0025 mg/mL to 0.02 mg/mL, the ability to remove DPPH free radicals increases. Compared with EBF, the scavenging ability of vitamin C for DPPH free radicals was stable, and when vitamin C was increased by 0.2 mg/mL, the scavenging activity of DPPH free radicals was the highest. The ability of EBF to clear DPPH was significantly lower than that of vitamin C (*p* < 0.05). Therefore, the scavenging ability of EBF to DPPH free radicals is weaker than that of vitamin C.

Through oxygen active oxidation, colorless ABTS are converted to stable blue-green ABTS • +, which is specifically absorbed at wavelengths of 734 nm. It has been widely used to evaluate the antioxidant activity of biological samples [39]. The scavenging ability of EBF on ABTS free radicals is shown in Figure 3D. The activity of EBF against ABTS free radicals is related to the amount of EBF, and the activity increases with the increase in EBF content. At the dose of 0.025 mg/mL, the scavenging activity of EBF is close to 50%. In addition, the activity of EBF against ABTS free radicals was significantly lower than that of vitamin C at all doses (*p* < 0.05). The results of the ABTS free radical decolorization test showed that the scavenging ability of EBF was weaker than that of vitamin C.

### 2.4. Effects of Epimedium Stem Flavonoids on Body Weight and Organs in CTX Mice

CTX is used clinically as an anticancer and immunosuppressant agent [14]. When accumulated in large amounts, it can cause oxidative stress in multiple organs, increase the production of reactive oxygen species (ROS), and damage cellular proteins and nucleic acids, thereby triggering apoptosis [40]. Degradation of antioxidant molecules in liver and kidney tissues caused in severe cases [17] occurs when acrolein and phosphoramidon bind to sulfhydryl groups in proteins, thus interfering with amino acid metabolism and leading to dysfunction [6].

In this study, the protective effect of EBF on CTX-induced kidney injury was investigated. It was found that the administration of CTX (80 mg/kg) resulted in significant weight loss, facial edema, and poor appetite in mice. MDA is elevated in serum and kidney, and impaired antioxidant system leads to kidney injury.

The results of EBF on the body weight, kidney, and kidney index of CTX-induced kidney injury mice are shown in Table 4. Compared with the blank group, the body weight of mice in the model group was significantly different (*p* < 0.01). CTX can significantly reduce the body weight of mice with kidney injury, and the results are statistically different. Compared with the model group, the kidney and kidney index of EBF dose group were significantly increased (*p* < 0.05), indicating that total flavonoids of stems and leaves of Herba fuli could improve kidney index of mice with kidney injury.

Observation of the appearance of the organs in Figure 4 showed that the kidneys and spleens of normal mice were healthy and red after dissection. In contrast, the kidneys of mice in the CTX group were partly whitish and swollen. The spleens were more petite, all relieved by EBF administration, with the kidneys gradually regaining their redness and the spleens increasing in size.

### 2.5. Effects of Biochemical Indices of Kidney Injury

MDA content can directly reflect the degree of body injury [41]. Oxidative stress is positively correlated with MDA levels [42]. The effect of the AMPK/mTOR signaling pathway on the protection of cordyceps sinensis against acute lung injury induced by acute kidney injury showed that MDA and BUN levels increased after renal injury (*p* < 0.01) [43]. In our study, as shown in Figure 4A,B, MDA content in serum and kidney tissue of the model group was significantly increased compared with the blank group (*p* < 0.01), indicating that the CTX-induced kidney injury model was successful. Compared with the model group, MDA content in serum and kidney of mice after EBF administration at different doses was significantly decreased (*p* < 0.05, *p* < 0.01), and was significantly decreased at 100 mg/kg EBF dose. In the study of renal fibrosis in which Icariin alleviates chronic kidney disease by inhibiting the activation of renal fibroblasts mediated by interleukin-1β/transforming growth factor-β, it was also found that Icariin, an effective component of the total flavones of Icariin, can significantly block the MDA content caused by kidney injury [44]. Based on the above results, it is suggested that a certain dose of EBF can significantly reduce the increased level of MDA in the kidney injury induced by cyclophosphamide in mice and has the ability to restore the antioxidant level of the body.

Clinically, the concomitant nephrotoxicity and hepatotoxicity caused by CTX limit its therapeutic effects, and CTX induces inflammation by increasing the activities of tumor necrosis factor-α (TNF-α), nuclear factor-κB (NF-κB), interleukin-6 (IL-6), and inducible nitric oxide synthase (INOS) [45]. On the other hand, changes in serum BUN levels can reflect the kidneys’ functional status and excretory function, which can also assess renal function [46].

Urea nitrogen is a metabolic end product of protein in the body and is an indicator of glomerular filtration function, visually reflecting renal function. The serum levels of blood urea nitrogen (BUN) and creatinine (CRE) were significantly higher in Wistar albino rats in the treatment of cisplatin-induced renal injury [47]. BUN and MDA levels were further increased in the study of kidney injury induced by septicemia in rats [48]. We detected the urea nitrogen content in the serum of mice. The results in Figure 4C show that compared with the blank control group, the serum BUN level in the model group was increased (*p* < 0.05), indicating that CTX modeling caused corresponding damage to the kidney function of mice. At the same time, compared with the model group, the BUN content in serum of mice after different doses of EBF was significantly decreased (*p* < 0.05, *p* < 0.01), of which 100 mg/kg EBF dose had the best effect (*p* < 0.01). In our study, BUN levels were significantly elevated in the CTX-treated group of mice, indicating that CTX is nephrotoxic. The elevation of BUN levels after CTX injection may be due to the alteration of cellular membrane permeability and body-circulation permeability after renal injury [49]. After BEF treatment, the BUN levels gradually returned to normal, indicating that the BEF gavage treatment effectively attenuated the CTX-induced nephrotoxicity. Plasma creatinine is a marker of kidney function, so we measured serum Cre [50]. The results in Figure 4D showed that compared with the control group, the content of serum renal function monitoring index (Cre) in the model group was significantly increased (*p* < 0.01), suggesting that CTX could cause kidney damage in mice. Compared with the model CTX group, different doses of EBF all decreased the level of Cre in mice, and there was a significant difference (*p* < 0.05, *p* < 0.01). The level of EBF in the 100 mg/kg EBF group decreased significantly (*p* < 0.01), suggesting that EBF has a preventive and protective effect on CTX-induced kidney injury. In the study on Icariin’s improvement of sepsis mortality and acute kidney injury, it was found that blood urea nitrogen and creatinine levels and proinflammatory cytokines levels of mice in the model group were increased to varying degrees, and ICA treatment significantly improved these renal changes, especially in the 60 mg/kg ICA group [51].

ALT and AST are the most widely used biochemical indicators to reflect hepatocellular injury in clinical practice, suggesting signals of liver disease [52]. The liver is an extremely important organ in the human body, and the damage of the kidney will affect the liver to maintain the balance of glucose in the body [53] as well as detoxification and metabolism of exogenous substances. Because elevated ALT levels are a marker of impaired liver function [54], we examined GPT/ALT levels in the liver. Liver and kidney damage leads to elevated serum AST, ALT, BUN, and creatinine levels [55]. As shown in Figure 4E, ALT/GPT levels in the liver of the model group were increased compared with the blank control group (*p* < 0.05), indicating that CTX modeling caused corresponding damage to the liver function of mice. At the same time, compared with the model group, different doses of EBF all decreased the ALT/GPT level in the liver of mice and had a significant effect (*p* < 0.05, *p* < 0.01), and the dosage of 100 mg/kg EBF was significant (*p* < 0.01). In the study of the protective effect of dietary Icariin on acute oxidative stress induced by lipopolysaccharide and hepatopancreas injury of eriocheir sinensis, it was found that ICA, an active substance in natural herbs, may be used as a feed additive because it can enhance the antioxidant capacity and non-specific immune function of animals. It is suggested that ICA can reduce liver injury by reducing the release of ALT and AST in the blood [56]. The results showed that EBF could reduce the hepatic enzymes in pathological mice and had a good protective effect on CTX-induced liver injury in mice.

IL-6 is an inflammatory factor and elevated levels can directly reflect the degree of body damage, suggesting that the body is undergoing an inflammatory response. Studies have shown that NF-κ B is closely related to renal diseases. NF-κ B is a transcription factor for many inflammatory factors and is essential to the inflammatory response [57]. NF-κ B can regulate the expression and production of pro-inflammatory cytokines and other inflammatory mediators. In addition, NF-κ B induces the expression of the inflammatory cytokines TNF-α, IL-1 β, and IL-6, amplifies the inflammatory cascade response, and is highly activated at some inflammatory disease sites [58]. As shown in Figure 4F–H, compared with the control group, IL-6 levels in serum, liver, and spleen of mice in the model group were significantly increased (*p* < 0.05) or (*p* < 0.01). The difference was significant in liver and spleen (*p* < 0.01), suggesting that CTX can affect the kidney-related liver and spleen injury in mice and increase the level of IL-6. Compared with the model CTX group, the levels of IL-6 in serum, liver, and spleen of mice were decreased at a 50 mg/kg EBF dose (*p* < 0.05), and the differences were significant in the 100 mg/kg EBF dose group (*p* < 0.01), suggesting that EBF can reduce the levels of IL-6 after CTX-induced kidney injury.

### 2.6. Expression Levels of Keap1 and Nrf2 Proteins in Renal Tissue of Mice in Each Group

Western blot results showed that, compared with the normal group, Keap1 expression in kidney tissue of the model CTX group was significantly increased (*p* < 0.01), while Nrf2 expression was significantly decreased (*p* < 0.01). Compared with the model group, Keap1 expression in renal tissues of EBF administration groups was significantly decreased at different doses (*p* < 0.05 or *p* < 0.01) and was significantly decreased at a 100 mg/kg EBF dose (Figure 4L). The expression of Nrf2 was significantly increased (Figure 4J) and was significant (*p* < 0.05 or *p* < 0.01). The results indicated that the Nrf2 signaling pathway could be regulated at 100 mg/kg EBF and the antioxidant capacity could be enhanced. Some studies have also found that the expression levels of p38 and p-p38 are also significantly increased in CTX-mediated AKI, suggesting that the mechanism of CTX may involve the activation of the p38 MAPK signaling pathway to induce apoptosis [59].

### 2.7. Histopathological Analysis

Histopathological changes are indicators for assessing the structural damage to the kidney, and the effect of CTX on renal histology has been confirmed in previous studies. It has been pointed out in the study on the protective effect of aminoguanidine on oxidative stress and kidney injury induced by cyclophosphamide in rats, that diffuse glomerulus and tubular nephritis [60], medullary tubule swelling, and lined epithelial cell hypertrophy can be observed in the renal cortex of rats treated with CTX [24].

The HE staining results of kidney tissue are shown in Figure 5A–H. In the normal group, the glomerular structure was normal, clear, the basement membrane was smooth, there was no inflammatory cell infiltration (Figure 5A), and the medullary structure was good (Figure 5E). Compared with the normal group, the glomerulus in model CTX group showed obvious atrophy and necrosis, renal tubular epithelial cell necrosis, glomerulus and renal tubular congestion (Figure 5B), chronic medullary inflammatory cell infiltration, and spontaneous calcification were often located at the junction of cortex and medulla (Figure 5F). Compared with the model CTX group, the kidney tissue damage of mice was reduced to varying degrees after administration of EBF at different doses. Glomerular necrosis was slightly relieved at the 50 mg/kg EBF dose (Figure 5C), and medullary infiltration was reduced (Figure 5G). In contrast, mice treated with a dose of 100 mg/kg EBF had significantly reduced renal tubule degeneration, tended to have normal glomeruli, reduced damage (Figure 5D), and good medullary condition (Figure 5H).

In addition, PAS staining results of renal tissue are shown in Figure 5I–P. In normal mice kidney structure integrity, there are glomerular morphology rules (Figure 5I) and a normal medulla structure (Figure 5M). Compared with normal mice, the glomerular area and mesangial matrix of the kidney in CTX group were enlarged (Figure 5J), and the medullary structure was affected (Figure 5N). However, with different doses of EBF, it was found that the kidney tissue injury of mice was reduced to varying degrees, and the glomerular area was slightly reduced at the dose of 50 mg/kg EBF (Figure 5K), and the medullary structure was relieved (Figure 5O). In contrast, the 100 mg/kg EBF dose group significantly reduced the surface area of the glomerulus, the expansion of the mesangial matrix of the glomerulus also decreased (Figure 5L), and the medullary structure normalized (Figure 5P). Therefore, histopathological results suggest that a high dose of EBF can reduce the damage caused by CTX on mouse kidneys.

### 2.8. Molecular Docking Validation of Core Targets and Active Compounds

In order to verify the accuracy of the target proteins, we screened the core target proteins of EBF on renal injury, KEAP1, NRF2, COX2, NFκB, and Icariin for molecular docking validation using Auto Dock Vina software 1.2.3. KEAP1 is a cytoplasmic protein that can act as a Cullin-3-based ubiquitin E3 ligase system junction that regulates various protein degradation, including NRF2, P62, and SQSTM1 [61]. Cyclooxygenase (COX) is the rate-limiting enzyme for converting arachidonic acid to prostaglandin-like and has two isoforms, i.e., COX1 and COX2, with 65% amino acid homology [62]. COX2 is expressed explicitly in the renal medulla [63], which helps maintain renal excretion, and impaired homeostatic imbalance of the renal medulla produces pathology after kidney injury. As shown in Figure 6, Finally, we used the Vina score (kcal/mol) to indicate the binding affinity between the target protein and the compound. The lower the score, the more stable the binding of the ligand to the receptor and the more likely the molecular interaction.

Our results show that Icariin binds well to the antioxidant target proteins KEAP1 and NRF2 and the anti-inflammatory target proteins COX-2 and NF-кB. As shown in Figure 6A,B, the graphs are based on the docking results of Icariin with KEAP1 and NRF2 pathways, and Figure 6C,D are the results of docking with COX-2 and NF-кB. The free binding energies of the docking with KEAP1, NRF2, COX-2, and NF-кB were −9.8 kcal/mol, −11.0 kcal/mol, −10.0 kcal /mol, and −8.1 kcal/mol, respectively. Icariin has been shown to significantly enhance the expression of Nrf-2 and heme oxygenase (HO-1) while attenuating the expression levels of Keap1, NF-кB, and COX-2 groups. This study illustrates the anti-inflammatory properties of Icariin [64]. As a fraction of molecular docking, binding energies less than −5 kcal/mol indicate more vigorous binding activity.

The Keap1-Nrf2 pathway plays an important role in maintaining body homeostasis. Nrf2 plays a crucial role in the basal and induced expression of antioxidant genes, which can counteract oxidative stress [65]. In previous studies, Icariin, the main active component of EBF, was found to protect the kidney from NLRP3 activation through the Keap1-Nrf2/HO-1 pathway and maintain the integrity of the glomerular filtration barrier [66].The results of this study showed that the levels of MDA, BUN, Cre, and IL-6 in serum, MDA in kidney, GPT/ALT and IL-6 in liver, and IL-6 in spleen were analyzed in each group of mice. It was found that 50 mg/kg EBF and 100 mg/kg EBF can reduce the levels of MDA, BUN, Cre, GPT/ALT, and IL-6 induced by CTX, suggesting that EBF can alleviate the oxidative stress, renal function injury, and inflammation caused by CTX. Western blot detection of Keap1 and Nrf2 protein expression in kidney tissue showed that Keap1 protein expression in model CTX group was higher than that in the blank group and EBF treatment administration group, and Nrf2 protein expression was lower than that in the blank group and EBF treatment administration group. It is suggested that total flavonoids of Epimedium can activate the Keap1-Nrf2 signaling pathway to reduce inflammation and oxidative stress of renal tissue, thereby alleviating kidney injury and improving renal function.

## 3. Materials and Methods

### 3.1. Materials

Cyclophosphamide is purchased from Yuanye Biotechnology Co., LTD., Shanghai, China. The stem and leaf pieces of Epimedium are purchased by the laboratory, and the certificate samples are stored in the laboratory together. Rutin, macroporous resin AB-8, anhydrous ethanol, 95% ethanol, normal saline, dimethylsulfoxide, polyamide,2, 2-biphenyl-1-picrobutyrohydrazine (DPPH), Vitamin C, the Malonaldehyde (MDA) kit, and GPT/ALT kit were purchased from Solaibao Biotechnology Co., LTD., Beijing, China. The interleukin-6 test kit was purchased from Huangshi Scientific Research Biotechnology Co., LTD., Nanjing, China. The BUN kit and creatinine kit were purchased from Suzhou Keming Biology Co., LTD., Suzhou, China. Instruments: Grinding machine, centrifuge, enzyme marker, electronic scale, KH-300DB380 CNC ultrasonic cleaning machine, mass spectrometry (MS), rotary evaporator, and water bath constant temperature oscillator were purchased from Shanghai Shensheng Technology Co., LTD., hanghai, China.

### 3.2. Optimization of the Extraction Process of Total Flavonoids from Epimedium Stems and Leaves

#### 3.2.1. Extraction of Total flavonoids from Epimedium Stems and Leaves

Precisely weigh 10 g Epimedium stems and leaves powder. Select ethanol concentration, material-liquid ratio, and ultrasonic time as the three influencing factors for experimental design. The total flavonoids content of the extract is the judgment criterion. Process ultrasonically three times and then combine the filtrate, rotary evaporation, and concentration.

#### 3.2.2. Standard Curve and Sample Determination

The rutin assay method was chosen to determine the total flavonoid content [67]. First, 5.0 mg of rutin standard was precisely weighed in a 50 mL volumetric flask and dissolved with 70% ethanol, and the volume was fixed to the scale. Next, 1.0 mL, 2.0 mL, 3.0 mL, 4.0 mL, and 5.0 mL of rutin standard solution was extracted to five 10 mL volumetric flasks. Then, 0.3 mL 5% NaNO_2_ solution was added, shaken well, and left to stand for 6 min; then 0.3 mL 10% A I(NO_3_)_3_ solution was added, shaken well, and left to stand for 6 min; then 4 mL 4% NaOH solution was added, and left to stand for 15 min; then 0.3 mL 10% A I(NO_3_)_3_ solution was added, shaken well, and then 4 mL 4% NaOH solution was added. Then, 0.3 mL 5% NaNO_2_ solution was added, shaken well, and left for 6 min; then 0.3 mL 10% A I(NO_3_)_3_ solution was added, shaken well, and left for 6 min; then 4 mL 4% NaOH solution was added and left for 15 min. Next, the volume was fixed by adding 70% ethanol to the scale to replace the standard solution of rutin with the same volume of 70% ethanol as the blank, and then the same method was repeated to determine the absorbance at 505 nm. The absorbance was measured at 505 nm. The standard curve was plotted with the concentration of rutin (c) as the horizontal coordinate and the absorbance (A) as the vertical coordinate [68].

#### 3.2.3. One-Way Experiment on the Extraction of Flavonoids from Epimedium Stems and Leaves

During the one-way test, the effects of volume fraction of ethanol (50, 60, 70, 80, 90%), different liquid/feed ratios (1:10, 1:15, 1:20, 1:25, 1:30 g/mL), and ultrasonic time (10, 15, 20, 25, 30 min) on the extraction rate of total flavonoids of Epimedium stems and leaves were investigated. The total flavonoids of Epimedium were extracted according to the above method, and the optimum process conditions were selected.

#### 3.2.4. Response Surface Experiments on the Extraction of Flavonoids from Epimedium Stems and Leaves

A four-factor, three-level response surface experiment was designed based on a one-factor experiment with ethanol concentration, material-liquid ratio, and extraction time as variables and total flavonoid content as response value using Box–Behnken design software of Design-Expert 8.0.6, and the results were subjected to regression fitting model and analysis of variance (ANOVA). The experimental design of 3-factor response surface analysis is shown in Table 5.

#### 3.2.5. Verification Experiment of Optimal Extraction Conditions

Response surface optimization was operating to obtain the optimal process for extracting total flavonoids from Epimedium, and the total flavonoid content was demonstrated under the optimal extraction conditions after parallel extraction. In this study, the best extraction process was to extract the total flavonoids of Epimedium for spare parts.

### 3.3. Purification and Mass Spectrometry (MS) Analysis of Flavonoids from Epimedium Stems and Leaves

The crude extract of total flavonoids from Epimedium was concentrated and dried using a rotary evaporator for later use. The AB-8 macroporous resin was weighed and soaked overnight in a 95% ethanol solution. The AB-8 macroporous resin had a significant trend of adsorption and resolving ability in the enrichment of flavonoid constituents compared with the other types of resins, so the AB-8 macroporous resin was used in this experiment [69]. The optimal purification process was carried out by using the extract (containing 15% ethanol) with the mass concentration of raw drug (0.5 g/mL) on the column, and the sample was taken at a flow rate of 1 BV/h for three hours at 25 °C. The water-soluble substances, such as sugars, were washed away with distilled water at 8 BV. Then, 25% ethanol at 5 BV was used for removal of other impurities and 60% ethanol was used at 4 BV for elution. The eluates were collected [70], and purified total flavonoids of Epimedium were obtained after vacuum drying. The purity of its total flavonoids reached 44.16%. Efficient analysis and detection of total flavonoid complexes in stem and leaf Epimedium was determined by chromatography-mass spectrometry [71].

### 3.4. In Vitro Antioxidant Activity Assay

Free radicals are metabolic products of various oxidative reactions in human life activities, which in excess can cause various damages to body cells, tissues, and organs, accelerate the aging process of the organism, and induce various diseases. Vitamin C (VC) is a potent antioxidant that can prevent body peroxidation and fight against harmful free radicals as a positive control [72].

DPPH free radical scavenging assay: In this experiment, the activity of DPPH radicals of Epimedium flavonoids was determined, and two radicals with purple-red characteristic absorption peaks were formed in DPPH-containing organic solvents [73]. Throughout the reaction, hydrogen atoms and electrons provided by the antioxidant combine with the DPPH radicals to form DPPH2 [74], changing the absorbance values and characteristic absorption peaks of the solution, which decrease and become smaller. The stronger the antioxidant capacity of the measured substance, the more the color faded. Instructions: Protect from light, weigh a certain amount of DPPH precisely, use ethanol as solvent, and prepare a reserve solution of 5.0 mmol/L. Pipette 4.5 mL of DPPH accurately. Accurately remove 4.5 mL of DPPH standard solution in a 10 mL cuvette, dispense to form a concentration gradient, set up a blank (A0) without sample solution, and then use anhydrous ethanol as solvent to make the volume to the scale line at 5.0 mL. Shake well, leave it to stand at room temperature for 0.5 h, and then determine the A517 nm value of the solution [75]. The clearance is then calculated, and the clearance is calculated according to the following formula:Clearance rate = (A0 − A1)/A0 × 100 per cent
where A1 is the absorbance of the test group, A0 is the absorbance of the control group.

ABTS^+^ free radical scavenging experiments:

The production of ABTS^+^ free radicals requires the oxidation of potassium persulfate, which has a maximum absorption at 734 nm, so their concentration can be determined by measuring the absorbance at 734 nm [76]. If the absorbance of a substance decreases after it is added to a solution of ABTS^+^ free radicals, the substance has free radical scavenging activity and is an antioxidant. A particular volume of sample extract was placed in a 10 mL cuvette, 4.9 mL of ABTS^+^ was added to form a gradient concentration, no sample was added to set a blank, and the volume was set to a water bath and heated (30 °C) with oscillation and shaking. The absorbance D1 was measured at room temperature and protected from light for 10 min. Meanwhile [77], a control was made to measure the absorbance D0, and the scavenging rate was calculated:Clearance rate = (D0 − D1)/D0 × 100 per cent(1)
where: D1 is the absorbance of the test group; D0 is the absorbance of the control group.

### 3.5. Animal Research

In this experiment, the 40 male Kunming mice with a body mass (of 18 ± 22) g were provided by Changchun Yise Laboratory Animal Technology Co. LTD., Changchun City, Jilin Province, China. and acclimatized for one week. All mice were housed in standard humidity (60 ± 5%) and room temperature (23 ± 2 °C) alternating between day and night for 12 h and given average food and water. The research was reviewed by the Experimental Animal Welfare and Ethics Committee of Jilin Agricultural University, and it was considered that the research project met the ethical needs of experimental animals. Ethics review acceptance number: 20211011003. (Experimental Animal Center, Jilin Agricultural University Experimental Animal License: SYXK (Ji) 2018-0023)

They were divided into four groups:blank control group (normal saline)model group (80 mg/kg cyclophosphamide intraperitoneal injection)total flavonoid low dose (50 mg/kg) grouptotal flavonoid high dose (100 mg/kg) group

Forty mice were randomly divided into four groups: Control group, model group (CTX80 mg/kg) [1], low dose (EBF, 50 mg/kg) in administration group [78], and high dose (EBF, 100 mg/kg) in administration group (Figure 7B). Except the control group, the other groups were intraperitoneally injected with CTX (80 mg/kg, dissolved in normal saline) for 7 consecutive days. Starting from day 8, low-dose and high-dose groups were given intragastric administration for 8 consecutive days, and the model group was injected with equal volume of normal saline. Mice in the control group were injected with equal volume normal saline once a day for 15 days. After the treatment, blood was taken from the eyeball, and supernatant −80 °C was centrifuged and stored in the refrigerator for later use. The mice were euthanized via neck dislocation. The whole kidneys were removed and weighed, some of which were used for subsequent biochemical and pathological experiments. Kidney index is calculated as follows: Kidney index (mg/g) = kidney weight/body weight.

#### 3.5.1. Sample Collection

In this study, we recorded the weight of the mice, collected the kidneys and weighed them, and calculated the kidney index. The blood was centrifuged to obtain the supernatant, which is used for biochemical analysis; Part of the fresh kidney tissue was immobilized with paraformaldehyde for histopathological analysis, and the other part was stored at −80 °C for western blot. Protein content was determined according to the instructions of the BCA kit (Solarbio, Beijing, China).

#### 3.5.2. Mouse Serum and Kidney Tissue Biochemical Assays

The serum levels of IL-6, MDA, Cre, and BUN were determined according to the instructions of the ELISA kit. After weighing the mice, blood was collected from the retro-orbital vein and dropped into a 1.5 mL centrifuge tube to separate the serum at high speed.

According to the instructions of MDA kit, MDA content in serum samples and MDA content in kidney tissues were determined. IL-6 kit (Huangshi Scientific Research Biotechnology Co., LTD., Beijing, China.) was used to determine the levels of IL-6 in serum, liver, and spleen. ALT/GPT kit (Solaibao Biotechnology Co., LTD., Shanghai, China.) was used to determine the level of alanine aminotransferase in liver. BUN and creatinine content in serum were measured by BUN kit (Suzhou Keming Biotechnology Co., LTD., Suzhou, China.) and creatinine kit (Nanjing Jiengcheng Biotechnology Co., LTD., Nanjing, China.) according to the instructions.

#### 3.5.3. The Protein Levels of Keap1 and Nrf2 in Kidney of Mice Were Detected by Western Blot

The supernatant was removed from the homogenate of mouse kidney tissue after cracking; the protein concentration was measured by BCA and denatured by boiling water bath after adding buffer. The suitable separation glue and concentrated glue were prepared according to the molecular weight of the protein. After adding TEMED, the glue was shaken and filled immediately. The sample was electrophoresis, and PVDF was transferred when bromophenol blue ran out of the suitable position. The membrane was closed with 5% skim milk in a shaker at room temperature and incubated overnight with primary antibody (Keap1 1:1500, Nrf2 1:1000) at 4 °C. The mixed ECL solution was added to fully react to start the exposure, develop and fix the image, and finally analyze the optical density value using the Image J software processing system [79].

#### 3.5.4. Histopathological Observations of the Mouse Kidney

Histopathological changes are an indicator for assessing structural damage to the kidney [12], and the effect of CTX on renal histology has been demonstrated in relevant studies [13]. Fixed kidney tissues were paraffin-embedded, and five-μm-thick sections were taken. H&E staining and PAS staining were performed according to standard protocols and observed under an light microscope.

### 3.6. Molecular Docking

EBF is the main pharmacologically active ingredient of the traditional Chinese medicine Epimedium, and its main active ingredient is the flavonoid constituent Icariin, which has been reported to have a wide range of pharmacological potential and has been demonstrated to possess anti-diabetic, anti-Alzheimer’s disease, antitumor, and hepatoprotective activities [14]. To validate the association between EBF and target proteins, we performed molecular docking simulations using its main component, Icariin, in AutoDockTools-1.5.6 software [80]. The crystal structures of the selected vital target proteins were downloaded from the Protein Data Bank in PDB format (http://www.rcsb.org/, accessed on 25 May 2023). The chemical structure of Icariin was obtained from the PubChem database (https://pubmed.ncbi.nlm.nih.gov/, accessed on 1 June 2023). The 3D structures of the active compounds were constructed by ChemBio3D Ultra 14.0.0.117 software while optimizing the energy minimization with the MM2 algorithm. Water molecules and organic compounds were removed from the receptor protein using PyMol software 2.5.5. [81]. The target protein receptor molecules were hydrogenated and charged by AutoDockTools-1.5.6 software [82], and the compounds and target protein receptors were converted to the PDBQT format. Finally, the molecular docking of potential targets and components was verified using the Auto Dock Vina software. Each set of molecular docking was performed at least three times, and the ionization energy was recorded. Visualization of docking results with optimal binding capacity was demonstrated using the BIOVIA Discovery Studio (2019) Visualizer [83].

### 3.7. Statistical Methods

The information obtained was statistically analyzed using SPSS 22.0 statistical software, and the information obtained was expressed as mean ± standard deviation. Statistical significance was analyzed using a one-way analysis of variance (ANOVA) followed by the least significant difference (LSD) provisional test. From a statistical point of view, a difference of *p* < 0.05 was considered significant.

## 4. Conclusions

In this study, three single factors, temperature, solid-liquid ratio, and ultrasonic time, were used to optimize the extraction process of total flavonoids from Epimedium. After optimization, the final extraction conditions were as follows: ethanol concentration of 60%, solid-liquid ratio of 25:1, ultrasonic time of about 25 min. The results of mass spectrometry showed that Icariin, Icariin A, Icariin B, and Icariin C were contained in the total flavonoid extract of Icariin stem and leaves, and Icariin was the best. In vitro DPPH and ABST+ tests showed that EBF had certain antioxidant activity in vitro. Animal experiments showed that 50 mg/kg EBF and 100 mg/kg EBF doses could reduce the increase in MDA, BUN, Cre, GPT/ALT, and IL-6 levels caused by CTX; Keap1 protein expression was decreased and Nrf2 protein expression was enhanced in renal tissue. Meanwhile, H&E and PAS staining were found to alleviate the histopathologic damage caused by CTX. Finally, Icariin was well bound to the antioxidant target proteins KEAP1 and NRF2 and the anti-inflammatory target proteins COX-2 and NF-κB by molecular docking. It is suggested that Icariin, the effective component of total flavones from stems and leaves of Epimedium optimized by extraction technology, may play a protective role in CTX-induced mouse kidney injury through the Keap1-Nrf2 pathway. This study lays a foundation for the subsequent study of Icariin in the treatment of kidney injury caused by broad-spectrum anti-tumor drug CTX, and also has certain reference help for screening natural drugs to prevent kidney injury induced by cyclophosphamide.

## Figures and Tables

**Figure 1 molecules-29-00207-f001:**
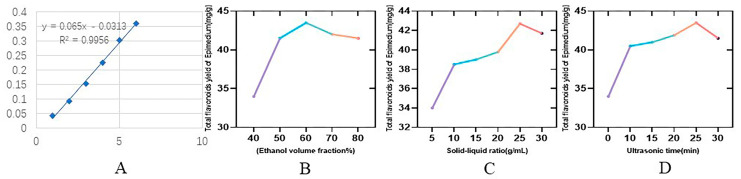
Standard curve of ruin and single factor results for extracting total flavonoids from stems and leaves of Epimedium. (**A**) Rutin’s standard curve; (**B**) Effect of ethanol volume fraction on total flavonoids yields of Epimedium; (**C**) Effect of solid−liquid ratio on total flavonoids yield of Epimedium; (**D**) Effect of ultrasonic time on total flavonoids yield of Epimedium.

**Figure 2 molecules-29-00207-f002:**
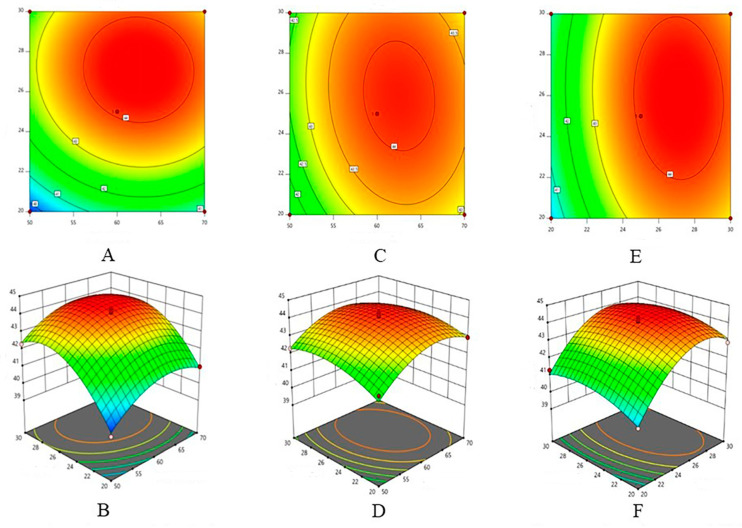
Experimental Design Results of Horizontal Response Surface Analysis for 3 Factors 3. (**A**,**B**) Response surface and contours of ethanol volume fraction and solid-liquid ratio; (**C**,**D**) Response surface plots and contours of ethanol volume fraction and ultrasonic time; (**E**,**F**) Response surface and contours of solid-liquid ratio and ultrasonic time.

**Figure 3 molecules-29-00207-f003:**
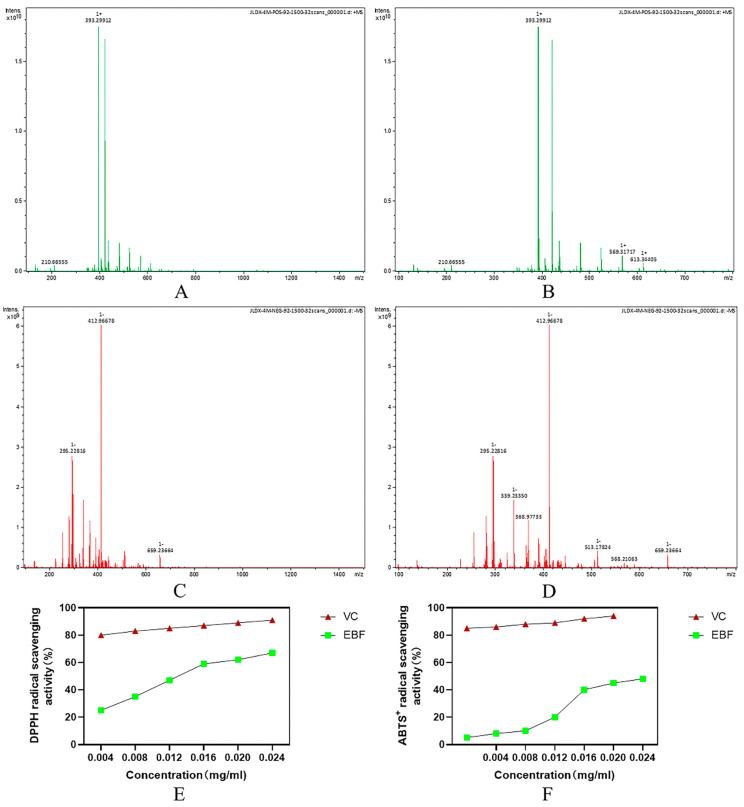
Mass spectrometry results and in vitro antioxidant capacity determination of purified total flavonoids from the stems and leaves of Epimedium. (**A**,**B**) Positive ion mode; (**C**,**D**) Negative ion mode; (**E**) DPPH radical scavenging ability; (**F**) ABST+ radical scavenging ability. (VC stands for vitamin C and EBF stands for Epimedium flavonoids.)

**Figure 4 molecules-29-00207-f004:**
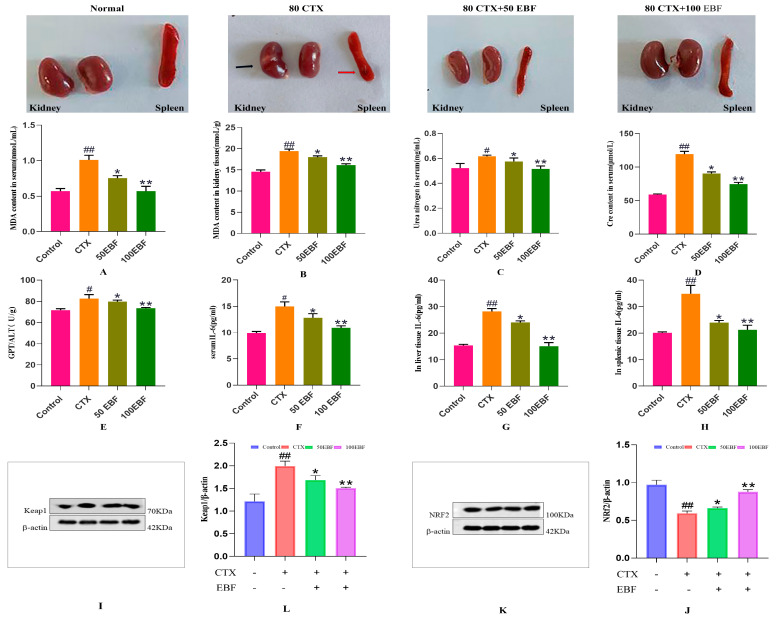
The first row shows photos of fresh kidneys and spleens in each group of mice. (**A**) MDA levels in serum of mice in each group; (**B**) MDA levels in renal tissues of mice in each group; (**C**) BUN content in serum of mice in each group; (**D**) Serum creatinine levels of mice in each group; (**E**) The level of alanine aminotransferase in liver tissue of mice in each group; (**F**) Serum IL-6 level of mice in each group; (**G**) IL-6 level in liver of mice in each group; (**H**) IL-6 levels in spleen of mice in each group; (**I**) Kidney Keap1 protein bands of mice in each group. (**L**) Expression level of Keap1 protein in kidney of mice in each group. (**K**) Nrf21 protein bands in kidney of mice in each group. (**J**) Expression level of Keap1 protein in kidney of mice in each group. Black arrow: kidney edema inflammatory lesions; Red arrow: splenic melanin deposits. Compared to the blank control group, # indicates a statistically significant difference between the control group and the model CTX group (# *p* < 0.05 or ## *p* < 0.01). Compared to the model CTX group, * represents a significant difference between the CTX group and the administration group EBF group (* *p* < 0.05 or ** *p* < 0.01). *n* = 10; Control: blank control group; CTX: Model group; 50 EBF: 50 mg/kg EBF administration group; 100 EBF: 100 mg/kg EBF administration group.

**Figure 5 molecules-29-00207-f005:**
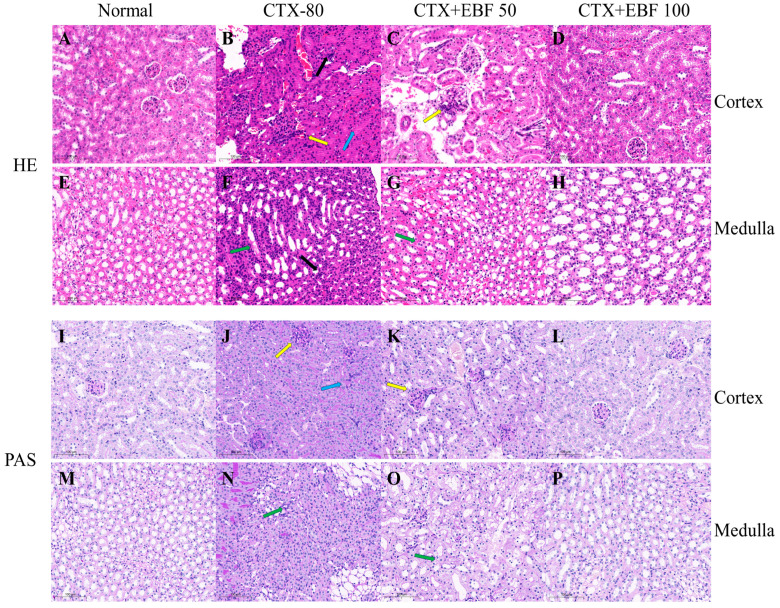
Histopathological examination of mouse kidney. HE and PAS control group showed normal renal tubule and glomerular structures (**A**,**I**), normal medullary structures (**E**,**M**). Renal tissue sections in model CTX group showed glomerular damage, tubular epithelial necrosis (**B**), medullary structural changes, and calcification (**F**). In the low−dose group (50 mg/kg EBF), the glomerular tubule injury was slightly reduced (**C**) and the renal medulla improved (**G**). The high dose group (100 mg/kg EBF) significantly improved the renal histomorphologic injury caused by CTX (**D**,**H**). Renal cortex and medullary PAS in the model group showed significant tubular necrosis accompanied by inflammatory infiltration (**J**) and medullary structural morphological changes (**N**). The low−dose group (50 mg/kg EBF) cortical PAS showed slight improvement in glomerular area and mesangial matrix enlargement (**K**), and slight improvement in medullary structural morphological changes (**O**). The high dose group (100 mg/kg EBF) showed that the glomerulonephric tubules tended to normalize (**L**) and the medullary structure tended to normalize (**P**). Microscope observation, scale: 100 microns. Yellow arrow: Increased glomerular area and mesangial matrix; Black arrow: inflammatory cell infiltration; Blue arrow: Tubular cell necrosis; Green arrow: Renal medulla structural morphological changes.

**Figure 6 molecules-29-00207-f006:**
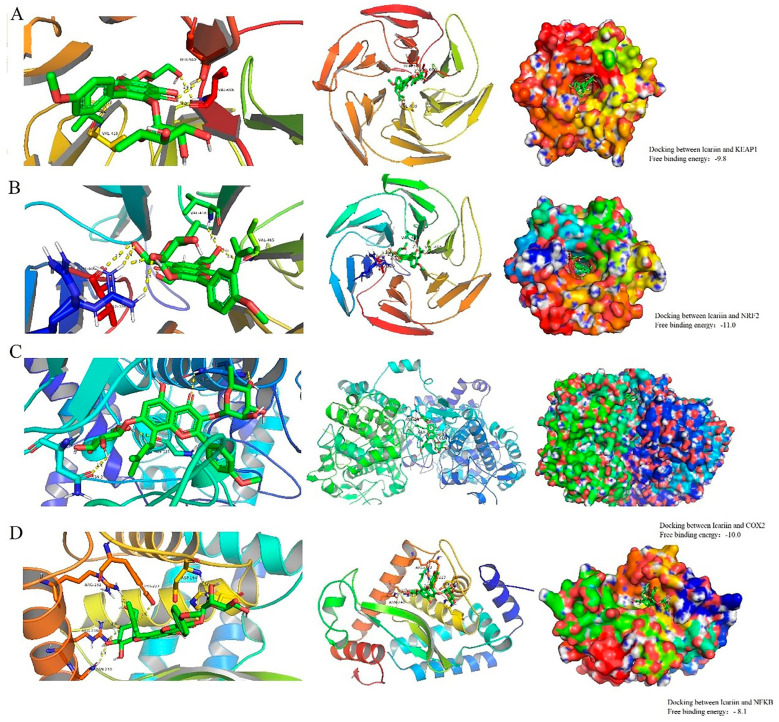
Molecular docking verification of core targets and active compounds. (**A**) Docking results of Icariin and KEAP1 target molecules, −9.8 kcal/mol; (**B**) Docking results of Icariin and NRF2 target molecules, −11.0 kcal/mol; (**C**) Docking results of Icariin and COX2 target molecules, −10.0 kcal /mol; (**D**) Docking results of Icariin and NFKB target molecules, −8.1 kcal/mol. (Combined free energy, also known as Gibbs free energy ΔG = −RTlnK, when the binding free energy value is negative, it indicates that the system is stable. If the free binding energy is less than −5, it indicates that the docking result is relatively stable, and the lower the better.).

**Figure 7 molecules-29-00207-f007:**
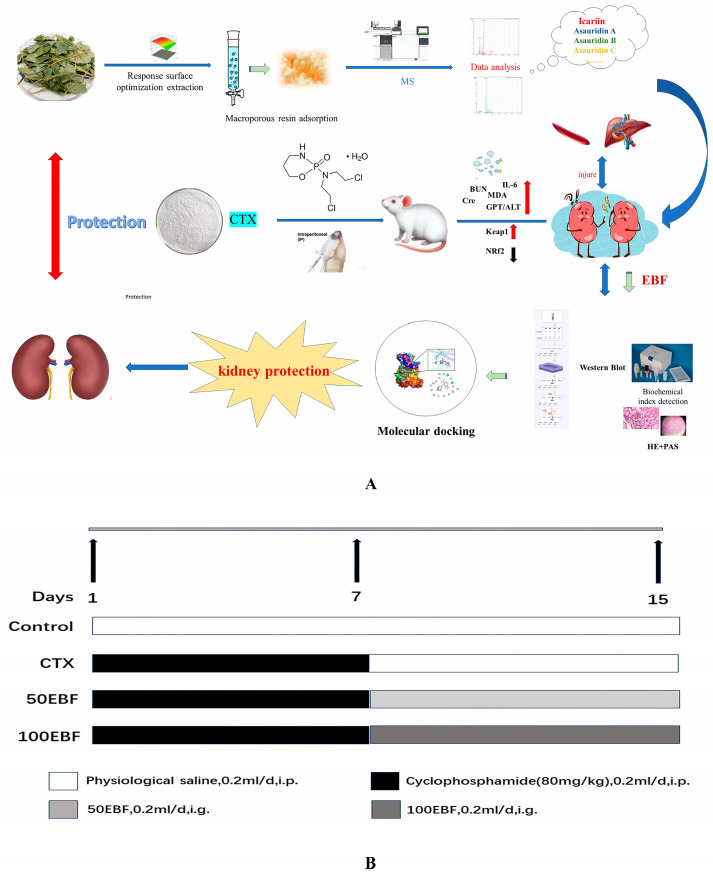
Design and Progress Schedule of Animal Research Experiments. (**A**) Flow chart of animal experiments in this study; (**B**) Flow chart of animal experiment design in this study; *n* = 10; Control: Normal control; CTX: Model group; 50EBF: low-dose group of total flavonoids from Epimedium; 100EBF:Epimedium total flavonoids high-dose group. ip: intraperitoneal injection, ig: gavage.

**Table 1 molecules-29-00207-t001:** Response Surface Analysis Scheme.

Test Number	X_1_ Ethanol Volume Fraction (%)	X_2_ Ratio (g/mL)	X_3_ Ultrasound Time (min)	Total Flavonoid Yield of Epimedium (mg/g)
1	−1	−1	0	39.4
2	1	−1	0	40.99
3	−1	1	0	42.3
4	1	1	0	43.5
5	−1	0	−1	41.8
6	1	0	−1	42.92
7	−1	0	1	42.3
8	1	0	1	42.8
9	0	−1	−1	40.3
10	0	1	−1	42.9
11	0	−1	1	41.3
12	0	−1	1	43.5
13	0	0	0	44.1
14	0	0	0	44.3
15	0	0	0	44.12
16	0	0	0	43.72
17	0	0	0	44.16

**Table 2 molecules-29-00207-t002:** Analysis of variance and significance test of regression model.

Source (of Information etc.)	Square Sum (e.g., Equation of Squares)	(Number of) Degrees of Freedom (Physics)	Mean Square	F-Value	PR > F
modelling	33.03	9	3.67	43.54	<0.0001
X_1_	2.43	1	2.43	28.84	0.0010
X_2_	13.03	1	13.03	154.58	<0.0001
X_3_	0.4900	1	0.4900	5.81	0.0467
X X_12_	0.0380	1	0.0380	0.4511	0.5234
X X_13_	0.0961	1	0.0961	1.14	0.3211
X X_23_	0.0400	1	0.0400	0.4745	0.5131
X_1_^2^	4.54	1	4.54	53.90	0.0002
X_2_^2^	9.39	1	9.39	111.45	<0.0001
X_3_^2^	1.45	1	1.45	17.17	0.0043
residual	0.5901	7	0.0843		
F-value of the mismatch ratio	0.4037	3	0.1346	2.89	0.1661
pure error	0.1864	4	0.0466		
aggregate	33.62	16			

**Table 3 molecules-29-00207-t003:** Mass Spectrometry Analysis Results of Total Flavonoids in Epimedium Stem and Leaf.

*m*/*z*	Search	PPM	Ion Types	Molecular Formula	Calculate Mass	Intensity	Chinese Name
367.1193	367.1187	1.63	[M-H]-	C_21_H_20_O_6_	368.1260	360345120	Isodehydro Icaritin or Anhydrocaritin or Anhydrocicaritin
403.0962	403.0954	2.01	[M+Cl]-	C_21_H_20_O_6_	368.1260	280661824	Isodehydro Icaritin or Anhydrocaritin or Anhydrocicaritin
407.0713	407.0722	−2.12	[M-2H+41K]-	C_21_H_20_O_6_	368.1260	912871	Isodehydro Icaritin or Anhydrocaritin or Anhydrocicaritin
413.1245	413.1242	0.84	[M+HCOOH-H]-	C_21_H_20_O_6_	368.1260	2518385	Isodehydro Icaritin or Anhydrocaritin or Anhydrocicaritin
427.1399	427.1398	0.16	[M+CH_3_COOH-H]-	C_21_H_20_O_6_	368.1260	5993696	Isodehydro Icaritin or Anhydrocaritin or Anhydrocicaritin
735.2453	735.2447	0.76	[2M-H]-	C_21_H_20_O_6_	368.1260	3180218	Isodehydro Icaritin or Anhydrocaritin or Anhydrocicaritin
535.1381	535.1376	0.81	[M+Cl]-	C_26_H_28_O_10_	500.1682	1107085	Icariin II
545.1666	545.1665	0.24	[M+HCOOH-H]-	C_26_H_28_O_10_	500.1682	1046836	Icariin II
493.1502	493.1504	−0.44	[M-H_20_-H]-	C_27_H_28_O_10_	512.1682	12936170	Icariin
557.1674	557.1665	1.69	[M+HCOOH-H]-	C_27_H_28_O_10_	512.1682	1417257	Icariin
571.1828	571.1821	1.24	[M+CH_3_COOH-H]-	C_27_H_28_O_10_	512.1682	2500022	Icariin
495.1662	495.1661	0.25	[M-H_20_-H]-	C_27_H_30_O_10_	514.1839	1322234	Icariin II
549.1534	549.1533	0.26	[M+Cl]-	C_27_H_30_O_10_	514.1839	2988083	Icariin II
573.1985	573.1978	1.33	[M+CH_3_COOH-H]-	C_27_H_30_O_10_	514.1839	1115969	Icariin II
511.1612	511.1610	0.39	[M-H_20_-H]-	C_27_H_30_O_11_	530.1788	1931379	Icariin I
565.1486	565.1482	0.63	[M+Cl]-	C_27_H_30_O_11_	530.1788	3307556	Icariin I
613.1927	613.1927	0.11	[M-H20-H]-	C_31_H_36_O_14_	632.2105	1870759	Icariin F
631.2031	631.2032	−0.19	[M-H]-	C_31_H_36_O_14_	632.2105	1578115	Icariin F
677.2100	677.2087	1.88	[M+HCOOH-H]-	C_31_H_36_O_14_	632.2105	1261814	Icariin F
645.2198	645.2189	1.47	[M-H]-	C_32_H_38_O_14_	646.2262	24605194	Arrowleaf Icariin B
681.1960	681.1956	0.59	[M+Cl]-	C_32_H_38_O_14_	646.2262	1442420	Arrowleaf Icariin B
691.2253	691.2244	1.29	[M+HCOOH-H]-	C_32_H_38_O_14_	646.2262	6129730	Arrowleaf Icariin B
695.2119	695.2112	1.01	[M+Cl]-	C_33_H_40_O_14_	660.2418	13050456	2″-Rhamnosylpastramonoside II
643.2039	643.2032	1.09	[M-H_20_-H]-	C_32_H_38_O_15_	662.2211	10209444	Icariin A
661.2145	661.2138	1.10	[M-H]-	C_32_H_38_O_15_	662.2211	4458815	Icariin A
697.1909	697.1905	0.63	[M+Cl]-	C_32_H_38_O_15_	662.2211	1738067	Icariin A
657.2198	657.2189	1.37	[M-H_20_-H]-	C_33_H_40_O_15_	676.2367	26417782	Icariin
675.2303	675.2294	1.24	[M-H]-	C_33_H_40_O_15_	676.2367	12396209	Icariin
697.2090	697.2108	−2.67	[M-2H+Na]-	C_33_H_40_O_15_	676.2367	4287317	Icariin
711.2068	711.2061	1.00	[M+Cl]-	C_33_H_40_O_15_	676.2367	14240036	Icariin
721.2360	721.2349	1.51	[M+HCOOH-H]-	C_33_H_40_O_15_	676.2367	1058330	Icariin
659.1989	659.1981	1.19	[M-H_20_-H]-	C_32_H_38_O_16_	678.2160	5541848	Hexandraside E
673.2147	673.2138	1.29	[M-H_20_-H]-	C_33_H_40_O_16_	692.2316	14542177	Icariin
713.2040	713.2058	−2.53	[M-2H+Na]-	C_33_H_40_O_16_	692.2316	5589455	Icariin
727.2019	727.2010	1.16	[M+Cl]-	C_33_H_40_O_16_	692.2316	10505580	Icariin
807.2720	807.2717	0.38	[M-H]-	C_38_H_48_O_19_	808.2790	1672456	Baohuoside V or Chaohuodin B
843.2497	843.2484	1.59	[M+Cl]-	C_38_H_48_O_19_	808.2790	2725050	Baohuoside V or Chaohuodin B
853.2779	853.2772	0.79	[M+HCOOH-H]-	C_38_H_48_O_19_	808.2790	961657	Baohuoside V or Chaohuodin B
821.2876	821.2874	0.28	[M-H]-	C_39_H_50_O_19_	822.2946	1491425	Chao Hao Ding C
857.2652	857.2640	1.40	[M+Cl]-	C_39_H_50_O_19_	822.2946	13225374	Chao Hao Ding C
859.2431	859.2427	0.49	[M-2H+K]-	C_39_H_50_O_19_	822.2946	1830197	Chao Hao Ding C
823.2677	823.2666	1.32	[M-H]-	C_38_H_48_O_20_	824.2739	3252587	Epimedium, genus of herbaceous flowering plant, cultivated in the Far East as aphrodisiac
819.2734	819.2717	2.08	[M-H_20_-H]-	C_39_H_50_O_20_	838.2895	1129049	Chao Hao Ding A
837.2841	837.2823	2.18	[M-H]-	C_39_H_50_O_20_	838.2895	2038814	Chao Hao Ding A
859.2618	859.2637	−2.12	[M-2H+Na]-	C_39_H_50_O_20_	838.2895	4975924	Chao Hao Ding A
873.2600	873.2589	1.20	[M+Cl]-	C_39_H_50_O_20_	838.2895	4745870	Chao Hao Ding A

**Table 4 molecules-29-00207-t004:** Changes in body mass and renal index of mice induced by cyclophosphamide in EBF.

Groups	Initial Weight (g)	Final Weight (g)	Kidney Weight (g)	Renal Index (mg/g)
Control	19.70 ± 1.23	27.32 ± 1.34	0.33 ± 0.05	1.50 ± 0.04
CTX	20.27 ± 1.01	23.23 ± 2.18 ^##^	0.30 ± 0.04	1.47 ± 0.03
50EBF	20.28 ± 1.56	25.18 ± 1.87	0.32 ± 0.04 *	1.48 ± 0.02 *
100EBF	20.70 ± 1.03	26.67 ± 1.01	0.33 ± 0.03 *	1.49 ± 0.02 *

The above table shows the changes in body weight and kidney and kidney index in mice. Compared with the blank control group, ## indicates a statistically significant difference between the control group and the model CTX group (## *p* < 0.01), and compared with the model CTX group, * indicates a significant difference between the CTX group and the drug administration group EBF group (* *p* < 0.05). *n* = 10; Control: blank control group; CTX: Model group; 50 EBF: 50 mg/kg EBF administration group; 100 EBF: 100 mg/kg EBF administration group.

**Table 5 molecules-29-00207-t005:** Experimental Design Table for Horizontal Response Surface Analysis of 3 Factors.

Level (of Achievement etc.)	X_1_ Ethanol Volume Fraction (%)	X_2_ Liquid Dose Ratio (g/mL)	X_3_ Ultrasound Time (min)
−1	50	1:20	20
0	60	1:25	25
1	70	1:30	30

## Data Availability

Research data used to support the results of this study are included in the article.

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
