# Peer review of "Optimization of the Flavonoid Extraction Process from the Stem and Leaves of Epimedium Brevicornum and Its Effects on Cyclophosphamide-Induced Renal Injury"

_molecules, 2023, doi:10.3390/molecules29010207_

Round 1
Reviewer 1 Report
Comments and Suggestions for Authors
Manuscript: Optimization of the flavonoid extraction process from the stem and leaves of Epimedium brevicornum and its effects on cyclo-phosphamide-induced renal injury
The manuscript to me is, in general, clearly written. The scientific and technical execution of the study is of good quality.
· Several linguistic and typo errors are obvious and must be revised carefully.
· Rephrase the conclusion in the abstract.
· The keywords were general, it is better to use more specified words reflecting the focus of your work
· Line 45; costly nd (please check nd)?
· Line 71: is the plant named (Epimedium et al.)?
· Line 93; add some recent citations on medicinal plants benefits used for cancer therapy:
- https://pubmed.ncbi.nlm.nih.gov/37440131/
· Figure 5 H&E staining of renal tissue; add arrows on the pathological lesions.
· Line 373: Why do you add a reference to the Rutin standard (Shanghai et al.)?
· Line 387; add a reference for extracting total flavonoids from Epimedium stems and leaves.
· In M&M; add references for the selected dose of cyclophosphamide.
· Line 518; add more details about blood sampling.
· Line 639; Please, follow the journal instructions in reference writing (2022 Sep;30(9):7355-7363.
· Line 643; Apr;33(4). e22271. doi: 10.1002/jbt.22271. Epub 2018 Dec 1. PMID: 30506662. ??
· All references must be revised according to journal guidelines in reference writing.
Comments on the Quality of English Language
The manuscript to me is, in general, clearly written. The scientific and technical execution of the study is of good quality.
· Several linguistic and typo errors are obvious and must be revised carefully.
Author Response
First of all, thank you very much for taking time out of your busy schedule to present this manuscript (ref. molecules-2658653) provide comprehensive guidance and professional comments to help us improve our articles. We sincerely thank you for your baby feedback, as you pointed out the relevant problems, we are very sorry for our carelessness, timely and carefully read all your comments, we have carefully corrected the manuscript to improve the quality of the manuscript. The revised part is highlighted in yellow. We submit the revised manuscript again. If there are any related problems in the manuscript, please do not hesitate to contact us in time, and we will revise it in time. The following is the revised reply after listening to your valuable comments:
- For key words: This study mainly explored the cyclophosphamide-induced kidney injury of epimedium stems and leaves after optimizing the extraction of total flavonoids through response surface experiment, combined with animal experiments. Icariin in the total flavones of icariin is the main active component of icariin. Therefore, in this study, we adopted a new technical method to simulate the interaction between small ligand molecules and receptor biomacromolecules - molecular docking. The new addition of icariin is generally specific.
- You point out 45 lines in the original draft: Change nd to AND
- You point to 93 lines in the original draft: The reference you recommended has been inserted
- Diseased arrows have been added to Figure 5
- In the original manuscript, you pointed out line 373: Adding Shanghai is the purchase address of Rutin, which has been deleted and modified
- In the original manuscript, you pointed out line 387: Relevant references have been added
- Relevant literature on cyclophosphamide dose selection has been added
- As for the references, they have all been added and modified according to the readjusted order of the journal guide
Finally, we sincerely thank you for your valuable advice.
Reviewer 2 Report
Comments and Suggestions for Authors
In the manuscript entitled "Optimization of the flavonoid extraction process from the stem and leaves of Epimedium brevicornum and its effects on cyclo-phosphamide-induced renal injury", the authors addressed two independent research objectives, the optimization of the flavonoid extraction protocol and the evaluation of their protective effect on tissues due to the action of CTX. Although this makes this study quite complex, the authors managed to design it in an appropriate way, making it intuitive and easy to follow. The applied research methodology is appropriate, which makes the results and conclusions reliable.
However, before final approval, I would have a few minor comments and suggestions that follow in the order in which I came across them:
Line 21, 168, 453, 583: In vitro – Italic
Line 23, 109: In vivo - Italic
Line 45: nd should be AND
Line 71: I believe that there is an error in the reference to the literature, considering that Epimedium is a type of plant and not the name of the author, especially since there is no reference in the bibliography whose author is Epimedium. Maybe the authors wanted to give the genus/species (Epimedium sp.) instead of the reference (Epimedium et al.).
Line 92: Why are the references listed in two separate brackets?
Line 120: There is some error in the part of the sentence - …is shown in Fig. 1, B in Fig. 1 shows that… Maybe it should be full stop after Fig. 1. And after that, The B part of the Fig. 1 shows that… Please check that.
Line 165: Is it possible to write molecular formulas in Table 3 with atomic numbers in the subscript? So it will be easier to notice the difference between the numbers (0) and the letters (O).
Line 183: …cleic acids. nucleic acids and nucleic acids, thereby triggering apoptosis [15-17]. [15-17].
Line 267: End of sentence or comma after “group of mice”.
Line 293: Figure 5 H&E staining of renal tissue shows… Rephrase. Haematoxylin and eosin staining (Figure 5) of renal tissue shows…
Line 307: Figure 5(full stop.) and bold
Line 348: Figure 6(full stop.) and bold
Line 349: Icariin (capital I)
Line 395: …for three h(ours), and delete ENTER between this and the next line
Line 511: Figure 7(full stop.) and bold
From the explanation of Figure 4 (Part of the legend: Using one-way ANOVA to analyze the data, the data showed that the average ± SD * of the experiment was compared with the blank control group * p<0.05, ** p<0.001, and *** p<0.0001; Compared with the model group (CTX group), # p<0.05, # # p<0.001, and ### p<0.0001 ), I cannot conclude what is shown in the graphics. The impossibility of distinguishing details due to poor image quality makes it even more difficult. If I am interpreting it correctly, I suggest you rephrase that part of the explanation as follows:
In the graphs shown, an asterisk indicates a statistically significant difference between the control and CTX groups (* p<0.05, ** p<0.001, and *** p<0.0001), while a bar indicates a statistically significant difference between the CTX and EBF groups (# p<0.05, # # p<0.001, and ### p<0.0001).
In addition, in the manuscript for the same control group you have three different expressions: control, normal control and blank control group. Since you described what the Blank control group is in the Animal research section, is it possible to explain that you will call that group Control in the future, so that you can then use it both in the text and on the graphics?
Lines 589/590: This section is not mandatory but may be added if there are patents resulting from the work reported in this manuscript. I don’t understand, which section is not mandatory? Conclusion? It is advisable to have a conclusion, especially considering that the discussion of the results ended with speculation.
Given that 40 mice were used in the study, I assume that an ethical clearance statement is required for the use of experimental animals. However, I did not notice such a statement in the manuscript.
In general, all images and graphics are displayed in poor resolution and it is necessary to improve this because in some cases it is impossible to see the details of what is displayed.
Author Response
First of all, thank you very much for taking time out of your busy schedule to present this manuscript (ref. molecules-2658653) provide comprehensive guidance and professional comments to help us improve our articles. We sincerely thank you for your baby feedback, as you pointed out the relevant problems, we are very sorry for our carelessness, timely and carefully read all your comments, we have carefully corrected the manuscript to improve the quality of the manuscript. The revised part is highlighted in yellow. We submit the revised manuscript again. If there are any related problems in the manuscript, please do not hesitate to contact us in time, and we will revise it in time. The following is the revised reply after listening to your valuable comments:
- You point out that lines 21,168,453,583 of the original manuscript have all been corrected in vitro - in italics, Line 23,109 is in - italics. 45 lines of nd correction AND
- In line 71 of the original manuscript, the problem with epimedium has been corrected
- The reference in line 92 of the original manuscript has been corrected due to my careless misrepresentation
- Lines 120, 267 of the original manuscript: The picture description problem has been modified by adding "."
- In line 297 of the original manuscript: the description of the HH staining picture has been modified according to your prompt, and the arrow of the lesion has been added. Each picture is marked with letters. Please refer to the revised manuscript for details
- In lines 307, 348, 349, 395 and 511 of the original manuscript: the bold text was corrected and ". "was added. Deleted Spaces.
- We have modified the problems you pointed out in Figure 4 of the manuscript according to your suggestions, and the image has been processed and upgraded with pixels. After modification, # represents the control group versus the model CTX group (#p< 0.05 or ##p<0.01). Compared with the model CTX group, * indicates that there is a statistically significant difference between CTX group and EBF group (*p< 0.05 or * *p< 0.01). Control: Normal control; CTX: Model group; 50EBF: low-dose group of total flavonoids from Epimedium; 100EBF:Epimedium total flavonoids high-dose group.
- Lines 589 and 590 in the original manuscript did not produce patents, due to our careless writing caused relevant errors, we have deleted this part after listening to your suggestion.
- We are very sorry for the animal ethics you mentioned. Now we have added an animal ethics statement to the manuscript to make up for the previous gaps. All images have been corrected with maximum effort.
Finally, we sincerely thank you for your valuable advice.
Reviewer 3 Report
Comments and Suggestions for Authors
Comments to the Author
1. What is novelty in the present study? The authors should present the effects of ETX treatment on specific antioxidant, anti-inflammatory, anti-apoptotic singling pathways in the kidney since the focus of this study was nephrotoxicity.
2. Title is too long and general; it should be more specific, including novelty of the study
3. Abstract needs to be rewritten because it contains many results which are not investigated and presented in the present study/manuscript! In line 24-24:” …and increase the level of SOD in the serum (P< 0.05) …” these results are not presented in the manuscript! Next, in line 25:” IL-6, MDA, and ALT in the liver tissue of mice…” There isn’t result of MDA level in the liver in the manuscript!
4. Introduction: give references for this statement “About 30% of this substance is excreted in the urine in the activated states, causing severe nephrotoxicity and urinary side effects in cystic fibrosis.” Give reference for: “…as well as other inflammatory markers.” in line 42. Give references for this statement:” Some natural products have been shown to alleviate CTX-induced nephrotoxicity during chemotherapy, but there is limited research on EBF.” – name natural products, studies, used doses of CTX that caused nephrotoxicity…
-in line 71 give a full name of a plant species, Epimedium brevicornum …
-give references for this statement:” Recent studies have shown that Epimedium can be used as a natural drug candidate for improved treatment and prevention of diabetes mellitus.” (in lines 72-74)
-paragraph in lines 107-111: the authors should clearly state what was the aim of the present study, and what were the targets of investigation regarding the effects of EBF treatment in animal model CTX-induced renal injury.
-the authors should clearly indicate novelty of the study. Also, it is essential to show that they induced nephrotoxicity in this model, and what were the most important effects of 50mg/kg and 100mg/kg EBF treatments, and were there any significant improvement in rats with CTX-induced renal injury.
5. Materials and Methods section has to be improved – regarding the Animal research, the authors should give the approvement for the experimental protocol and the permission number for the experiment from the Ethic Committee!
- Write which gender of mice was used, males or females (line 492)?
- Why did you use the dose of 80 mg/kg for CTX? Give the relevant references.
- How did you choose the doses for EBF (50 mg/kg, and 100 mg/kg), and length of the treatments, based on what?
- In lines 501-505 the authors described experimental design and wrote that the Epimedium total flavonoids will be injected intraperitoneally for seven consecutive days starting from day 4, but in Figure 7, under (B) one can clearly see that presented experimental design is not in accordance with this text; Namely, the authors presented that the treatments with 50EBF and 100EBF were given by gavage (“ig: gavage” in legend of Figure 7), and that these treatments started from day 7. And lasted 8 consecutive days!? Also, regarding CTX, in presented flow chart CTX were given for seven consecutive days, while in the text in line 502 it is said that CTX treatment lasted for three consecutive days!?
- In line 517 “Kidneys, spleen, liver, and other tissues were excised …”, the authors should name what “other tissues” they were collected in this study. Also, in sections “3.5.1. Sample collection” and “3.5.2. Mouse serum and kidney tissue biochemical assays” there are a lot of repeated parts, the authors should rewrite lines 517-538 in a clear way and unite these subsections into one.
- In lines 523-527: Firstly, the aim of the study shouldn’t be stated here in Materials and Methods, its place is at the end on Introduction section; Secondly, the authors stated that “This study aimed to investigate the protective effect of total flavonoids of Epimedium on cyclophosphamide-induced renal injury after optimizing the extraction process so the contents of nephrotoxicity indexes, antioxidant enzymes, malondialdehyde (MDA), inflammatory factors, NF-KB significant proteins, and apoptotic pathways were determined.” which clearly are not the results that are presented in this manuscript!; namely, there are no results of nephrotoxicity indexes, antioxidant enzymes, NF-KB significant proteins, and apoptotic pathway, and only one inflammatory factor (IL-6) is presented.
- In lines 536-538: how is it possible to measure blood urea nitrogen (BUN) in the tissue!?
- Histopathological examination: the authors should describe the histopathological scoring that was used for the assessment of histopathological changes in renal tissue (scoring of histologic lesions enables quantitative assessment and statistical analysis of treatment effects)- based on which results the authors compared the changes in renal tissue between the experimental groups? What type of microscope was used, and under which magnification the renal sections were examined?
- In lines 544-545: “It has been shown that EBF can improve CTX-induced renal injury.” Please give reference for this statement, and discuss it further in light of your results in Discussion section.
- In lines 570-571: Instead of this statement: “Statistically, differences were considered significant at p " 0.05, while p < 0.01 is considered particularly significant.”, it is enough to write “p-value <0.05 was considered significant.”
6. Results: The authors should pay a lot more attention to Results section that is loaded with unnecessary comments. Results should be revised in more comprehensive way, because it is very difficult to follow. The authors should present data obtained in the present study, and discussed them later in Discussion section.
-in subsection “2.3. In vitro antioxidant activity assay results” results regarding DPPH are completely missing.
- In legend of Figure 3. (lines 168-170) give what the abbreviations of VC, and EBF are stand for.
- In line 177: the authors said that “…the mass concentration of EBF is 0.020 mg/ml when the clearance reaches about 50%.”, but from the graph in D in Figure 3. One can clearly see that EBF concentration is 0.025 mg/ml.
- Based on which results the authors claim that the antioxidant system is impaired in the kidney, liver, and spleen (line 190)? Where are those results presented?
- In lines 365-368: “Combined with our previous tests of anti-inflammatory and anti-oxidative indicators in serum and related tissues in mice, we speculate that EBF may exert a protective effect against cyclophosphamide-induced renal injury in mice via the KEAP1-NRF2 oxidative-inflammatory pathway.” Please give references of these previous studies of yours, and name which antioxidant and anti-inflammatory tests were used there. Name what “related tissues” were analyzed at that time.
- In lines 187-188:” In the present study, we investigate the protective effect of EBF against CTX-induced kidney injury due to oxidative stress, inflammation and apoptosis.” Where are the results regarding apoptosis? They are missing from the manuscript.
- in line 190-191:” … and impaired antioxidant system in the kidneys, liver, and spleen.” The authors haven’t presented any results regarding antioxidant system in liver or spleen in this manuscript!
- in lines 194-200: there is a major discrepancy regarding the results presented in this paragraph and ones in Table 4. Furthermore, the authors should do again the statistical processing of the results! Also, the authors correct these mistakes regarding labeling the significance, in line 202: “** p<0.001”, while in line 206: “**p< 0.01”! In addition to this, in lines273-275 in Figure 4.:” … the data showed that the average ± SD * of the experiment was compared with the blank control group * p<0.05, ** p<0.001, and *** p<0.0001; Compared with the model group (CTX group), # p<0.05, # # p<0.001, and ### p<0.0001, …” Please correct these also! Proper labeling should be: *p<0.05, **p<0.01, ***p<0.001!
- In lines 214-216: “… in our study, EBF effectively reduced the MDA content and increased the activity of antioxidant enzymes in the kidney [23].” Firstly, the authors haven’t presented any results of antioxidant enzymes in the manuscript. Secondly, from this sentence one can understand that given reference “[23]” is the authors previous paper, but it is work of different authors and on different animal model (non-toxic renal fibrosis) and with different treatment protocol (The Sirt1 activator, SRT1720, attenuates renal fibrosis by inhibiting CTGF and oxidative stress. Yunzhuo Ren 1, Chunyang Du 1, Yonghong Shi 1, Jingying Wei 1, Haijiang Wu 1, Huixian Cui 2, Affiliations: 1Department of Pathology, Hebei Medical University, Shijiazhuang, Hebei 050017, P.R. China. 2Department of Anatomy, Hebei Medical University, Shijiazhuang, Hebei 050017, P.R. China.). The authors should present their results obtained in the present study, and discus them in Discussion section.
-in line 216: “As shown in Fig. 4 in F and C graphs, the MDA content in renal tissues and serum of mice…” These are not correct, in F is The Content of IL-6 in Mouse Spleen, and in C is Content of alanine aminotransferase in mouse liver. Pay more attention in labeling the graphs and correct all the other mistakes later on.
- in addition to BUN, it is essential to present the creatinine level in serum in order to assess kidney function in this CTX-induced renal injury.
-Regarding the assessment of nephrotoxicity, it is essential to present markers of inflammation in the kidney and impaired renal structure and function! Here there isn’t any, the authors are suggested to perform the western blots for NF-kB, IL-6, TNF-a, Keap1, Nrf-2, COX-2, and Masson’s trichrome staining of renal tissue.
-In lines 291-292:” Histopathological changes are indicators for assessing the structural damage to the kidney, and the effect of CTX on renal histology has been confirmed in previous studies.” Give references for this statement, name in which previous studies.
-In Figure 5, in legend label with different symbols (arrows, arrow head, ect.) all specific changes that you described/observed in renal tissue on micrographs and in the text/legend; also, add scale bars on every micrograph and in legend.
7. Discussion section is completely missing in the manuscript! The authors should write this section and discus all the obtained results from the present study.
8. The authors should also revise a Conclusion section: there is a discrepancy between the abstract, the aim and conclusions of the study. Furthermore, conclusions overstate the results and partly answer the study question.
9. References should be checked and corrected: The majority of the references are not presented in accordance with the guidelines/requirements of the article; furthermore, references [26], [27], [28], [29], [30], [31], and [32] are not cited in the manuscript; also, refence [31] is missing from this list as well. (Lines 696-709)
Comments on the Quality of English LanguagePay more attention on spelling.
Author Response
First of all, thank you very much for taking time out of your busy schedule to present this manuscript (ref. molecules-2658653) provide comprehensive guidance and professional comments to help us improve our articles. We sincerely thank you for your baby feedback, as you pointed out the relevant problems, we are very sorry for our carelessness, timely and carefully read all your comments, we have carefully corrected the manuscript to improve the quality of the manuscript. The revised part is highlighted in yellow. We submit the revised manuscript again. If there are any related problems in the manuscript, please do not hesitate to contact us in time, and we will revise it in time. The following is the revised reply after listening to your valuable comments:
- Questions about the novelty of this study: The novelty of this study is mainly based on the response surface optimization of the total flavonoids in the stems and leaves of Epimedium and the simulation of the interaction between ligand small molecules and receptor biomacromolecules by MS analysis、animal experiments have shown that icariin, an effective component in the total flavonoids of epimedium stems and leaves, may reduce renal tissue inflammation and oxidative stress by activating Keap1-Nrf2 signaling pathway, thus alleviating kidney injury.
- First of all, thank you very much for your valuable suggestions. In this study, the optimization of extraction was involved. Although molecular docking was done, the proportion was small and the depth was not achieved. After discussion, the problem was not corrected for the time being
- With regard to the abstract, we have listened to your valuable suggestions and have revised and rewritten the abstract as a whole. After this revision, the original research results of the manuscript are equally reflected. As for the MDA level in the liver, we are very sorry. Due to the negligence of the manuscript, the MDA level in the liver is too high in this manuscript, so the result of this sentence has been deleted in the manuscript.
- Introduction: After listening to your valuable comments, we have revised the relevant issues in this section by adding "... References to nephrotoxicity and urinary side effects; In line 42 of the original manuscript, references have been provided to "some natural products have been shown to reduce CTX-induced nephrotoxicity during chemotherapy, but studies on EBF have been limited." Line 71 of the original manuscript gives the full name of the plant species of Epimedium. It also provides references to "Epimedium can be used to improve the treatment and prevention of diabetes" 15. You pointed out the problems in lines 107-111 of the original manuscript, and we have made relevant modifications according to your suggestion. Modified as follows:“……, The levels of BUN, Cre, MDA, IL-6 in blood and MDA in renal tissue were higher than those in the control group, suggesting that CTX caused oxidative stress, renal function injury and inflammation. The results showed that both 50mg/kg and 100mg/kg EBF could reduce hair removal and facial edema in CTX model mice. Biochemical indexes showed that serum and liver and spleen inflammatory factor IL-6 levels increased, serum and kidney MDA levels increased, serum BUN, Cre levels increased, pathological tissue damage increased……”
- The section on materials and methods has been rewritten and corrected. The approval and permit number of the Animal Ethics Committee for this study was added.
- In line 492 of the original manuscript, we timely corrected the gender of the mouse, male, and corresponding references were also provided for CTX modeling dose. In the selection of dosage and time of administration, we investigated the relevant literature, which has been added later. As for the experimental design described in the original manuscript 501-505, we must apologize for our carelessness and carelessness here. As you can see, it is marked on our experimental design drawing that the whole experiment was conducted for a total of 15 days. However, we made serious mistakes in the process of description and translation. After your careful guidance and prompt, we made timely corrections as follows: … Except the control group, the other groups were intraperitoneally injected with CTX (80mg/kg, dissolved in normal saline) for 7 consecutive days. Starting from day 8, the low - and high-dose EBF groups were given intragastric administration for 8 consecutive days, and the model group was injected with equal amounts of normal saline. Mice in the control group were injected with normal saline once a day for 15 consecutive days…
- As you mentioned, there is overlap between sample collection and biochemical determination. We have made timely modifications to reduce duplication and make the description more specific. In response to your statement that the purpose of the study should not appear in the method, we made the relevant deletion. For the renal toxicity indicators you mentioned, we have remeasured the creatinine (Cre) level in serum and the problem you mentioned about how to determine BUN. We obtained blood samples by taking blood from eyeballs during sampling and obtained supernatant by centrifugation after sampling. Test according to the instructions,BUN kit and creatinine kit were purchased from Suzhou Keming Biology Co.
- As for the histopathologic sections you mentioned, we added lesion arrows, compared the changes among the renal tissues in each group, and observed the stained sections using biological microscope ML31. Multiplier x100. he relevant results are discussed in the original manuscript 544-545. In the original manuscript, 570-571, the issue of salience has been deleted in accordance with your proposal.
- As for the unnecessary results you mentioned, we have made timely deletion and modification and discussed them. In vitro antioxidant results have been timely supplemented (As the EBF concentration increases from 0.0025 mg/mL to 0.02 mg/mL…….) In Figure 3, the meanings of VC and EBF are marked. There are errors in the expression of line 177 in the original manuscript, and we have made timely adjustments in the current manuscript. Thank you for your careful guidance and suggestions. In the original manuscript, the description of the damage of the antioxidant system of the kidney, spleen and liver appeared to be exaggerated. We feel very sorry for that. MDA indexes were measured in the kidney, IL-6 levels only in the spleen, and markers GPT/ALT and IL-6 levels were measured in the liver. At the same time, we measured the expression levels of Keap1 and Nrf2 proteins in kidney tissue
- As for the apoptosis mentioned in lines 187-188 of the original manuscript, we are very sorry for the error in writing here. Apoptosis was not involved in our study, so we have deleted this part. As for lines 194-200 of the original manuscript, there are major errors between the results in this paragraph and those in Table 4. Thank you very much for your careful guidance, and we will correct this part as follows: Compared with blank group, the weight of mice in model group was significantly different (P<0.01). CTX can significantly reduce the body weight of mice with kidney injury. Compared with model group, the kidney and kidney index of EBF groups with different doses were increased (P<0.05). # indicates a statistically significant difference between the control group and the model CTX group (##p<0.01), * indicates a significant difference between the CTX group and the drug administration group EBF group (*p<0.05). n = 10; As for the wrong expression of antioxidant enzymes in the manuscript, we have made relevant deletion in time. In part 214-216 of the original manuscript, we have added the relevant references and discussed this part. In Figure 4, there was an error in the marking of graphic letters, which we corrected in time and will pay more attention to in the future. Thank you again for your careful guidance.
- As you pointed out, creatinine (Cre) should be added to evaluate kidney function. In the new manuscript, we have listened to your suggestion to increase the determination of serum Cre level. As for the relevant assessment of renal function, we are very grateful for your suggestion. Since the revision time of the manuscript was only ten days, we only added the protein expression of Keap1 and Nrf2 in renal tissue. For Masson staining, we are very sorry that we can only provide HE and PAS double staining. In lines 291-292 of the original manuscript, "Histopathology is an indicator for the assessment of damage to the kidney structure, and the effect of CTX on the kidney tissue has been confirmed in previous studies", the reference is added and indicated for previous studies.
- In line 297 of the original manuscript: the description of the HH staining picture has been modified according to your prompt, and the arrow of the lesion has been added. Each picture is marked with letters. As for the histopathologic sections you mentioned, we added lesion arrows, compared the changes among the renal tissues in each group, and observed the stained sections using biological microscope ML31. Multiplier x100.
- As for the discussion part, thank you very much for your guidance. Originally, we put the result and discussion together in the process of editing the manuscript, but due to our negligence, we forgot to add the title of discussion to the title of the result, and the discussion has been added to the revised manuscript. In response to your questions about the relationship between the summary, purpose and conclusion of the manuscript, we would like to express our gratitude. With your suggestions, we have made a new statement about the summary, purpose and conclusion. And removed the original language that should not be irrelevant.
- As for the reference you mentioned in the original manuscript, we are very sorry for the carelessness in editing the manuscript. The original references were added manually, the format was not uniform, and the position was confused, so some references could not be added in time. We actively listen to your suggestions, and in the revised manuscript, we reinserted all of them using Endnote software in accordance with the requirements of the article reference guide.
Round 2
Reviewer 1 Report
Comments and Suggestions for Authors
- The reference number 29; the year of publication is 2021, not 2020, and the volume, issue, and page numbers are wrong. please correct to
Sallam AO, Rizk HA, Emam MA, Fadl SE, Abdelhiee EY, Khater H, Elkomy A, and Aboubakr M, 2021. The Ameliorative Effects of L-carnitine against Cisplatin-induced Gonadal Toxicity in Rats. Pak Vet J, 41(1): 147-151.
Thanks
Reviewer 3 Report
Comments and Suggestions for Authors
The authors acknowledge all comments.
In line 655: a typo, "BAC" correct to "BCA"